# Forest Inventories in Private and Protected Areas of Paraguay

Andrew G. Cervantes *, Patricia T. Vega Gutierrez and Seri C. Robinson

Department of Wood Science & Engineering, Oregon State University, Corvallis, OR 97331, USA;
patricia.vega@oregonstate.edu (P.T.V.G.); seri.robinson@oregonstate.edu (S.C.R.)
* Correspondence: acerva02@gmail.com; Tel.: +1-267-205-3722

**Abstract:** Environmental degradation in Paraguayan ecosystems relates to anthropogenic intervention and has intensified with wildfires and drought. To help mitigate forest cover loss, the local government created laws, parceled land to private owners, and established protected areas. How differences in land tenure regimes affect dendrology and biodiversity remain to be answered. This preliminary study examined the differences and similarities between three protected area forests (National Park Ybycuí, and Natural Monuments Cerro Koi, Cerro Chorori) and three family-owned forests in Guairá, Central, and Paraguarí of eastern Paraguay. Forty-three transects were established following FAO protocols to examine tree diversity, floristic composition, and complementary vegetation structure information using two biodiversity indexes. Nine hundred and three individuals were registered, representing 92 species, 80 genera, and 35 families. The most abundant families were Fabaceae, Rutaceae, Myrtaceae, and Rhamnaceae; and the most abundant genera were *Pilocarpus, Columbrina, Acrocomia, Plina*, and *Astronium*. The most important species were *Parapiptadenia rigida, Peltophorum dubium*, and *Astronium fraxinifolium*. Results showed higher species richness and significantly greater tree diversity, but a nonsignificantly larger diameter in private forests compared to protected areas studied. Detected differences were small, indicating a need for additional research of forests with differing land tenure regimes as climate dynamics continually shift and management strategies show effects.

**Keywords:** tree diversity; biodiversity; floristic composition; protected areas; Paraguay

## 1. Introduction and Background

### 1.1. Biological, Climatic and Landscape Characterization of the Studied Area

Tropical forests are home to over half of the Earth's biodiversity and are major regulators and influencers of the global climate system via water transpiration, cloud formation, and atmospheric circulation [1]. Beyond being a habitat for innumerable living beings, forests offer the foundation of life through ecological services and serve as spaces for cultivation, materials, recreation, community engagement, spiritual enrichment, and more [2–4]. Not surprisingly, human population dynamics combined with developmental pressures and average climatic events have led to major planetary health challenges, such as environmental degradation, biodiversity loss, socioeconomic inequalities, and adverse conditions for human communities in the tropics, developing countries, and in some regions of South America [5,6].

Paraguay is located on the Tropic of Cancer (23°26′13.8″ S) and contains an area of approximately 408,000 km², Figure 1 [7]. Over 7 million inhabitants reside in 17 geopolitical departments throughout the country, with over 85% of the population in eastern Paraguay and nearly 4 million people residing in the Asunción Metropolitan Area (AMA) [7,8]. The country is officially bilingual, including the Guaraní and Spanish languages, and contains 19 indigenous populations from five different linguistic families [5,9]. The region has developed a rich ethnobotanical and traditional ecological knowledge base with the cultivation of *Ilex paraguariensis* St.-Hil. (Aquifoliaceae) and numerous other medicinal plants [10–12]. Thus, many nontimber forest products are commercialized, processed, and/or used daily as

medicine, as food/nutrition, for art as fibers, dyes, carvings, veterinary use, as aromatics, and in construction or fuel [3,5,7,8,11,13–18]. In addition to traditional ecological knowledge of Guaraní cultures, various nationals and foreigners have contributed to the understanding of dendrology and forest structure in Paraguay. Multiple herbariums exist in Paraguay, online databases have been established to support identification, books on plant taxonomy and identification keys are available, and various local scientific journals from the National University at Asunción publish studies on Paraguayan flora (*Rojasiana, Steviana, Ka'aguy*).

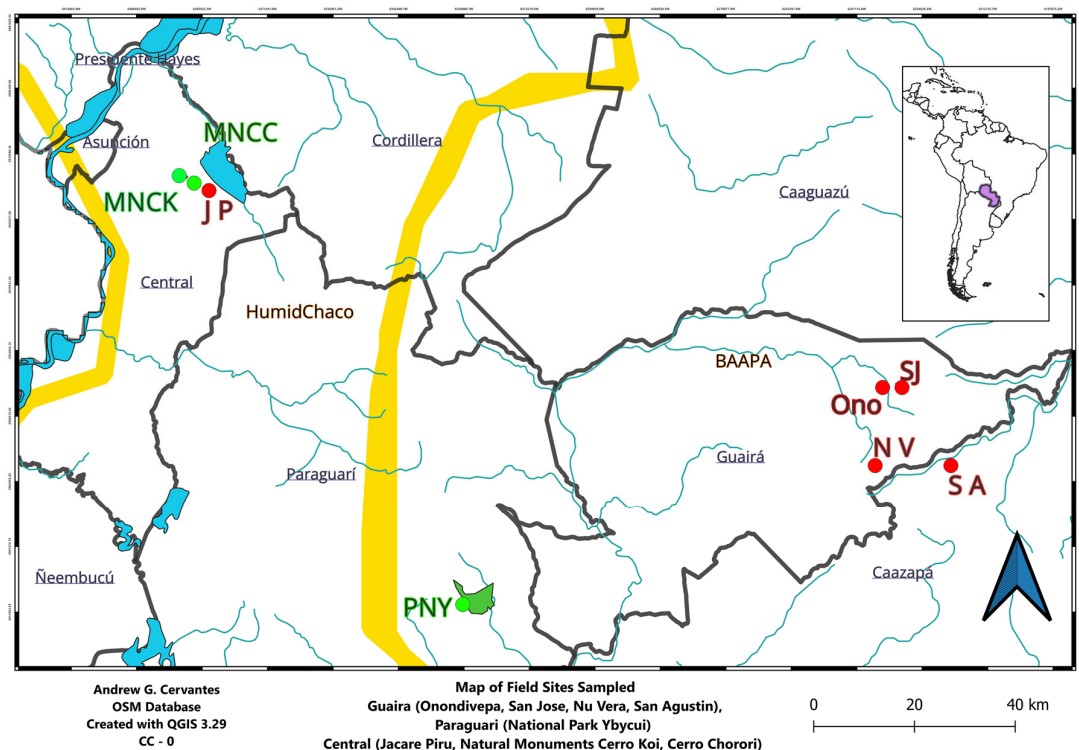

**Figure 1.** Map of field sites. Ecoregions (orange lines), departments, and field sites studied in eastern Paraguay. Private Property Forests (red markers): Guairá 1 (Ono, SJ), Guairá 2 (Nu Vera, San Agustin), Jacare Piru (JP). Protected Area Forests (green markers): Natural Monument Cerro Koi (MNCK), Natural Monument Cerro Chorori (MNCC), National Park Ybycuí (PNY).

The country includes multiple ecoregions that display consistencies in soil types, topography, and climates, as well as a range of flora, fauna, and fungi that extend across regional and international borders [5,19–23]. Western Paraguay shares portions of the Great American Chaco with Argentina and Bolivia and is the most extensive natural forest in South America after the Amazon. Here, ecoregions include the Pantanal wetlands, Humid Chaco, and Dry Chaco. Chaco vegetative formations include alluvial plains, savannahs, palm forests of the multipurpose *Copernicia alba* Morong., and emblematic meso-xerophytic semideciduous formations of tannin-rich *Aspidosperma quebracho-blanco*, Schldl., *Schinopsis balansae* Engl., and *Schinopsis quebracho-colorado* (Schldl.) F. Barkley & T. Meyer, and edible carob tree forests (*Prosopis* spp.). Encompassing 42% of national territory, the Dry Chaco contains grey sandy-clay soils and occasional lagoons with heavy salinization, and its average rainfall is the lowest in the country, <400–1000 mm/year, with temperatures averaging between 10–50 °C [5,23]. The Humid Chaco landscape is directly influenced by the Paraguay, Parana, and Pilcomayo River basins (approximately 32% of the national territory), characterized by flood-prone lowlands, average rainfall between 560–2080 mm/year, sandy soils, and temperatures between 0–49 °C [23,24]. As a convergence zone, the region supports a wide variety of arid and humid tolerant species. Meanwhile, eastern vegetative formations include Atlantic Forest (or BAAPA, for *Bosque Atlántico de Alto Paraná*), urban forests, humid semideciduous hardwood forests, riparian forests, wetlands, Cerrado

formations of savannas, tall forests, pastures, and secondary vegetation in degraded land-scapes [15,23]. The BAAPA (21% of national territory) is characterized by average rainfall between 790–3300 mm/year, temperatures of 24–40 °C, and soils of iron-rich red clay and basalt [5,17,23].

Biodiversity registries in Paraguay show approximately 1600 species of vertebrates, including: 500 fish, 85 amphibians, 140 reptiles, 170 mammals, 700 native bird species, and over 40 species of annual migratory birds [5,19,20,25–29]. Large fauna include jaguar, ocelots, tapir, monkeys, peccary, anteaters, deer, armadillo, guacamayos, rheas, and crocodiles, among others. Of the Paraguayan flora, estimates range between 6500–7000 total species of seed-bearing vascular plants (dicotyledons, monocotyledons) [30], that encompass 298 genera and 767 species of trees and bushes [30–32]. Other authors estimate up to 13,000 species, including spore-producing plants, nonvascular plants, and plant symbioses (Thallophyta, Bryophyta, and Pteridophyta) [33,34]. Today, over 80 vascular plant species are threatened/endangered throughout the country [5,35–37].

Although integral to planetary health via international biological corridors with other South American countries [38], environmental stability in eastern and western Paraguay are threatened by various concerns, which have continuing effects. Between 2010–2015, Paraguay was a country with the highest rate of forest loss globally, with more than 80% of the forest loss located in the Chaco [39–42]; despite being considered a priority area for conservation, only 9% of the Chaco is protected [1]. Similarly, only 13% of the original BAAPA forest remains, as there are degrading fragments across eastern Paraguay and Brazil with portions in protected areas [5,43–45], and both *Cerrado* and BAAPA ecoregions are considered biodiversity hotspots [38–41]. Major threats to ecosystems include pollution of inorganic waste, heavy metal soil contamination, air contamination, deforestation, pasture burning, and wetland draining practices, generally followed by unsustainable extractive land management systems and soil erosion [5]. Such repeated acts have influenced landscape transformations, including salinization, wildfires, habitat degradation, and changes in rain/drought patterns over time [5]. Beyond forest structure, the country faces questions of political autonomy, while integrating environmental justice and climate change mitigation programs as a developing country in a global capitalist economy.

### 1.2. Status of Forest Management Practices in Both Land Tenure Regimes

Land tenure and management are historically complex in Paraguay because of constant adaptation—from colonization, imperialism, liberation, political dictatorships, war, democracy, and ongoing corruption—resulting in unfair land distribution and interconnected social, environmental, economic, and educational injustices [46–49]. The governing bodies of protected areas are the Ministry of the Environment and Sustainable Development (MADES) and the National System of Protected Areas in Paraguay. Collectively, federal protected areas were established to conserve natural ecosystems [20,50–52]. Legislation has repeatedly been passed to stop deforestation in eastern Paraguay [53–55]. Between 1966 and 2007, over 50 federal and private protected areas emerged, encompassing 14.9% of the national territory [20,52]. However, federal agencies are understaffed, underequipped, and underfunded to comprehensively monitor the areas, and more regulation and infrastructure are needed to support the governance of environmental law in *Mbaracayu* Reserve, San Rafael Reserve, and *Reserva Natural Privada Tapytá* [55–62]. In response to the absence of educational material that supports the development of more resilient communities, internationally funded workshops and contemporary publications focused on Urban Forestry and Forest Restoration have taken place [63–68]. Here, management plans of the three PAFs studied were referenced [69–71]. In the National Park Ybycuí (PNY) management plan, a few studies listed offer sampling strategies and structural data. Two studies that focused on flora examined vegetation of rocky outcrops and Cactaceae distribution [72–75]. Within the management plan for Natural Monuments Cerro Koi and Cerro Chorori (MNCC, MNCK), only one vegetative study was published by Soria and Basulado (2004) [76]. This study highlights species found within protected areas yet does not include a sampling strategy

or data. In a related study using satellite imagery, Yanosky et al., (2009) showed that most protected areas conserved forest cover during the 1970–2000s; however, the buffer zones around protected areas experienced greater forest cover loss, signifying space for more cross-sectoral collaboration [77]. Field-based studies, coupled with satellite monitoring, are equally necessary for integral understandings of forest population dynamics [77,78].

In the private sector, private investors and the National Forest Institute (INFONA) support *Eucalyptus* spp. forest plantations to sustain the daily demand for fuel wood, and lumber and to reduce the dependency on remaining natural forests and protected areas [79,80]. Many rural forests under private ownership in Paraguay are less studied and documented in the scientific literature. By omitting rural forests from the literature, there is an incomplete understanding of the state of forests in the region and country. Hence, although in theory, management of protected areas and private forests may differ, the actual practice, or illegal effects of both, may render management differences moot. Thus, the repeated monitoring of forest population dynamics is essential to better understand the state of forest ecosystem services and natural resources that they produce over time. Both public and private sectors in Paraguay and globally have the capacity to regulate their existential–ecological balance, generate economic capital, conserve biodiversity, sustain production of natural resources, and mitigate climate change.

Comparative investigation and ongoing registries of vegetative populations between Private Property Forests (PPFs) and Protected Area Forests (PAFs) are limited and needed in Paraguay. This article aims to offer registries of dendrological inventories using field-based sampling methodologies and measurements in 43 transects divided into three PAFs: MNCK, MNCC, and PNY, as well as three PPFs in the departments of Guairá and Central. The objectives of this study were to raise biodiversity awareness in Paraguay and document similarities and differences in tree biodiversity via field-based dendrological-focused inventories in different land tenure regimes. Ongoing studies are necessary for broader understandings of the state of forests, so all parties and stakeholders can make collaborative, educated decisions on best practices for land management.

## 2. Materials and Methods

### 2.1. Description of Areas Studied

Eastern Paraguay extends from the Paraguay River to the Parana River border with Brazil and Argentina, with 14 of the country's departments and approximately 160,000 km$^2$ [7]. Study sites were in the departments of Central, Guairá, and Paraguarí. Guairá and Paraguarí are classified as BAAPA, and Central is Humid Chaco. Temperatures vary between 0–49 °C with wet seasons during the warmer months (October–April), dispersed over rolling hills and riparian lowlands with elevations between 46–842 m [23]. Access to PAF sites was available with written permission from MADES. Similarly, access to PPF sites required permission from landowners, established with help of fellow Peace Corps Volunteers and community contacts. Six different forests were sampled. A total of 43 transects were studied: 19 PPFs and 24 PAFs. Specific site information is found in Appendix A, Table A1.

The Central Department sites are located at elevations of 200 m above the *Ypacaraí* Lake Managed Resource Reserve. Nine transects were studied between May and July 2020 in the PPF Jacare Piru (JP) with a total area of 900 m$^2$/630,643 m$^2$, located 32 km from Asunción [71]. Three km from the JP PPF transects were the PAF sectors MNCC and MNCK. Both are of national value for the columnar joint sandstone geological formations seldom encountered on Earth [69,81]. The total area of MNCK is 12 ha, and that of MNCC is 5 ha, each lying two km apart and divided by private properties. In both, nine transects were inventoried between February and March 2021. Threats to the ecosystem around JP, MNCK, and MNCC include mining, forest extraction (generally as fuelwood for trade in the local ceramics industry), wetland conversion for cattle or monocultures, small settlements by land squatters, and large estates by landowners.

The six transects in PNY of the Paraguarí department were inventoried in July 2021, including a range of forest types, rocky outcrops, waterfalls and riparian vegetation, and gallery forests with a total area of over 5000 ha [70,74,75].

The Guairá sites were integral riparian forests and wetland landscapes. Four PPFs pertaining to four different communities of the *Paso Yobaí* district in rural Guairá were grouped into two different sites: Guairá 1 (G1) and Guairá 2 (G2), 169 km and 187 km from Asunción, respectively. Six G1 transects were inventoried in the communities of *San José* and *Oñondivepa,* three transects each, on 9 September and 4 October 2018, with a total area of 600 m$^2$/101,198 m$^2$. Similarly, six G2 transects were measured in the communities of *San Agustin* and *Ñu Vera,* three transects each, on 11 August and 10 November 2018, with a total area of 600 m$^2$/24,165 m$^2$.

*2.2. Methodology*

Methods for conducting forest inventories were based on protocols by FAO with some modifications [2,82]. For sampling, satellite maps were printed of each forest using Google Earth Pro [83]. Quadrants were drawn and divided equally, assigned a number and field sites were selected randomly using a random number generator, Random.org [84]. Limiting factors required each transect to be square in design and to measure 10 × 10 m 100 m$^2$ (0.01 ha), varying from the FAO recommendations (1 × 1 km or 60 × 60 m). Between three and nine transects were surveyed within each forest.

In the field, communication with local authorities on site conditions followed a safety assessment, in which, two randomly selected transects were replaced due to hazards. In almost all sites, a local guide, park guard, university students, or landowners accompanied the researchers to help take measurements and identify the vegetation. Forest descriptions and classifications were based on observations, cross references with the literature, data collected, and casual communication with local officials.

Data was recorded on inventory spreadsheets. The GPS coordinates and site height of the plot were taken in the southeast corner of the plot using a Garmin GPS etrex 3. Boundaries were measured using a Belota tape measure (10 m) and Spencer's Loggers Tape (100 ft) and were flagged every five meters with fluorescent colored tape. To determine if a variable tree was in or out of a transect, a 10-degree glass wedge rectangular prism was used following standard protocol [85,86].

Field measurements included: diameter at breast height (*dbh*), basal area, height, ground cover, taxonomical identification, and nonmeasurable data. Scale factors of transect size in reference to total forest area were calculated by summing transect total area and dividing by the total area of the forest; see Appendix A, Table A1. Photographic registries with an iPhone 6S (Apple, Cupertino, CA, USA) of leaves, flowers, seeds, or fruits of the vegetation were collected. Taxonomic data collected included: common name, species name, genera, and family. Nomenclature was based on cross references with established online databases: TROPICOS Paraguayan Project of the Missouri Botanical Garden [87], Instituto de Botánica Darwinion [88], and Flora do Brazil [89]. Tree identification was supported by the team, guidebooks, [11,13,15,30], local botanists, and regional herbariums. Unidentifiable samples were classified into the highest taxon. Here, trees measured were defined as woody perennial plants with one principal trunk, or in the case of understory shrubs, multiple stems with a defined crown. The *dbh* was calculated in cm, with circumference over bark at (1.3 m) for trees with circumference ≥ 10 cm. Individuals with circumferences <10 cm were identified and contributed to ground cover percentages. The *dbh* was found using:

$$d = \frac{C}{\pi}$$

where d is diameter, C is circumference, and $\pi$ is 3.14. Raw data was reported in metric units. Tree diameters were studied individually and later grouped into diameter classes of 0.10 m increments (0.01–0.80 m).

Basal area, the amount of space a trunk occupies, was measured to help describe stand density and biomass on a per hectare basis. Basal area was calculated by:

$$BA = \frac{\pi(dbh^2)}{4}$$

where *dbh* is in cm, and π is 3.14. Basal area data were presented in m units.

Vertical height (measured in m) of a tree from ground level to the uppermost leaf/branch of the crown was measured using a Suunto clinometer, measuring tape, and the formula:

$$H = (c \times d) + i$$

for measurements made on level ground, where H is tree height, d is the distance from base of the tree to the clinometer, c is the angle percentage of inclination from the clinometer to crown tip, and *i* is the height of the clinometer from the ground [90].

Ground cover is the percentage of material (vegetation, rocks, litter, moss, lichens, or bare ground) on the ground that can cover soil in a plot. Percentages were estimated by taking the average of all ground cover predictions by the researchers present within the transect. A value of 0% meant that the soil was bare, while a value of 50% meant a 50:50 mix of bare soil to vegetation, and a value of 100% meant that there was no exposed soil whatsoever.

The Important Value Index (IVI) highlights important species, as outlined by various authors [91–95]. The IVI was based on three sets of data: (1) relative frequency (the number of times that a species was encountered in a plot throughout the entire study), (2) the relative density/abundance (the number of times a species was recorded in all the plots within a criterion, and relative abundance is the abundance of a species relative to all other species), and (3) the relative dominance (absolute and relative basal area). Each data set was expressed as a percentage between 0–100%. IVI is the sum of all three and ranges from 0–300. Thus, species with higher IVI were considered "more important" to forest composition.

The Shannon–Wiener Diversity Index (SWDI) was utilized to offer information on site diversity in each transect. Shannon entropy quantified the uncertainty associated with the prediction of tree species [96–98], calculated by:

$$H' = -\Sigma \, pi \ln (pi)$$

where: Σ is the "sum," ln is the natural log, and *pi* is the proportion of the entire community made up of species, *i*.

### 2.3. Data Analysis

Of each sector evaluated in this study, there were three variables following protocols of [2,81]. The variables related to diversity were the number of individuals per ha, the number of species per sample site, and the number of families per transect. The values of the SWDI were analyzed with *t*-tests to compare diversity values of population mean to determine if diversity under different tenure was statistically significant, with α = 0.05. The variables related to floristic composition were families, genera, most abundant species, and rare species. The structural variables, in a preliminary and complementary fashion, included basic structural parameters. For *dbh* values, a *t*-test was run to determine if differences in mean *dbh* of forests with different land tenure regimes were statistically significant, with α = 0.05. Data were introduced into a database and abundance, basal area, frequency, diametric distribution, and the important value index were calculated [95].

## 3. Results

### 3.1. Dimensional and Taxonomic Comparisons between Protected and Private Forests

Nine hundred and three individuals were registered, representing 35 families, 82 genera, and 92 species. The most diverse botanical family across all land tenure regimes was Fabaceae with fourteen species, followed in decreasing order by Myrtaceae (eight species), Rutaceae (seven species), Sapindaceae (five species), and Lauraceae, Meliaceae, and Moraceae (four species each). Highly represented families registered in PPFs were Rutaceae, Rhamnaceae, and Sapotaceae families. While in PAFs, the most represented families were Arecaceae, Anacardaceae, and Myrtaceae, Figure 2.

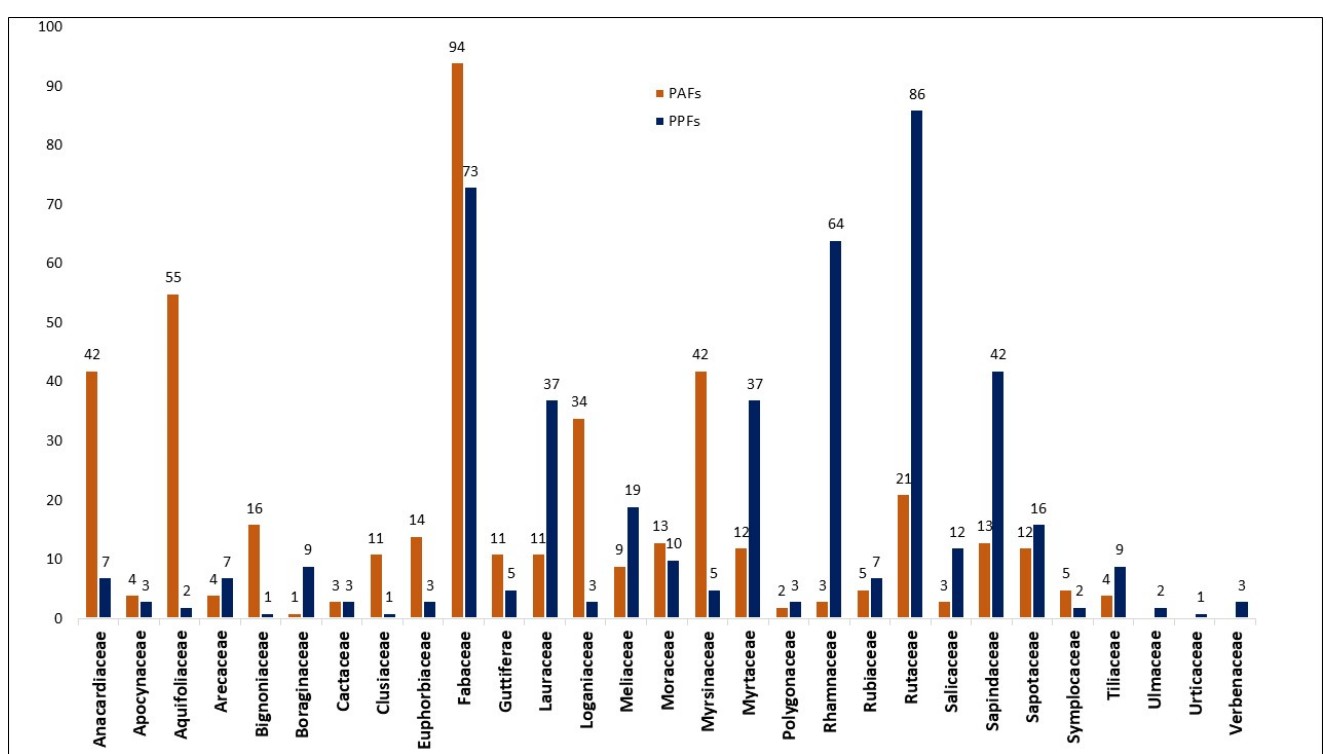

**Figure 2.** Plant families with the largest number of species in 19 PPF transects and 24 PAF transects of 0.01 ha in Humid Chaco and Atlantic Forests in Central, Guairá, and Paraguarí.

In this study, there was less species richness in PAFs than in PPFs. In the PAFs studied, 439 total samples representing 64 species were registered, while within PPFs, 464 total samples representing 72 species were recorded. Sample quantities across all sites are presented in Appendix A, Table A2. In the three PPF sites, the average number of species was 35. G1 registered 35 spp. in 191 samples, G2 registered 39 spp. (largest in the category) in 166 samples, and JP registered 32 spp. in 129 samples. Similarly, in three PAF sites, the average number of species was 33. MNCC registered 41 spp. (largest in the category) in 190 samples, MNCK registered 24 spp. in 104 samples, and PNY registered 34 spp. in 148 samples.

The average *dbh* for all samples was slightly less in the PAFs sampled (11.45 cm) in comparison to the PPFs sampled (12.40 cm). Results from *t*-tests indicate that *dbh* differences were not statistically significant. The average *dbh*s of PPFs were as follows: G1 was 9.8 cm, G2 was 12.8 cm, and JP was 11.1 cm. With measurements from outstanding individuals of *Albizia hassleri* (Chodat) Burkart (Fabaceae) measuring 47.7 cm in G1, *Nectrandra lanceolata* (Lauraceae) measuring 70 cm in G2, *Ruprechita laxiflora* Meisn. (Polygonaceae) 48.7 cm, *Cordia americana* L. (Boraginaceae) 47.4 cm, and *Parapiptadenia rigida* (Benth.) Brenan (Fabaceae) measuring 74.2 cm in JP. In comparison, average *dbh*s of PAFs were 5.92 cm in MNCC, 6.13 cm in MNCK, and 8.84 cm in PNY. Large individuals were *C. americana* (66.5 cm in PNY

and 60.5 cm in MNCC) and *Anadenanthera columbrina* (Vell.) Brenan (Fabaceae), 46.8 cm in MNCK. According to the diameter classes, in both private and protected areas observed in Figure 3, the greatest number of individuals were in the 0.01–0.09 m range.

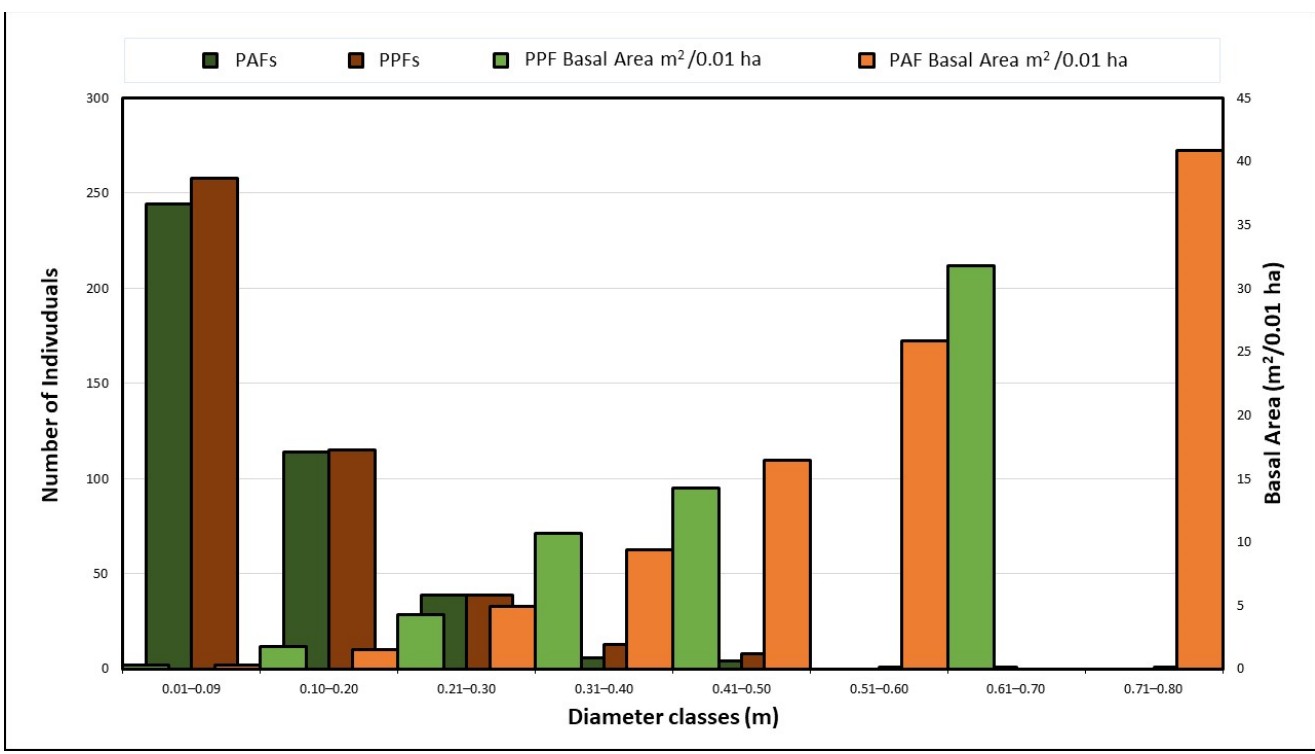

**Figure 3.** Diameter classes and basal area for all samples in PAFs and PPF.

*3.2. The Important Value Index and Shannon–Wiener Diversity Index*

The high-frequency species in PPFs were: G1: *Pilocarpus pennatifolius* Lemaire (Rutaceae); G2: *Columbrina retusa* var. latifolia (Rhamnaceae); and JP: *Plinia rivularis* Cambess (Myrtaceae) of the intermediate forest level. Across PPFs, the species with the greatest frequency were *Ocotea suaveolens* (Meisn.) Hassler (Lauraceae), *Lonchocarpus leucanthus* Burkart (Fabaceae), and *Chrysophyllum gonocarpum* (Mart. & Eichler) Engl. (Sapotaceae). In contrast, in MNCK and MNCC, the most frequent species was *Acrocomia aculeata* Mart. (Arecaceae), constituting palm-savannah formations, whereas in PNY, they were fruit bearing, *P. rivularis*, and the canopy level, *Copaifera langsdorfii* Desf. (Fabaceae). Across land tenure regimes, the most frequently measured species were *P. rigida*, *Astronium fraxinifolium* Schott (Anacardiaceae), *Peltophorum dubium* (Sprengel) Taubert (Fabaceae), and *C. americana*.

Absolute and relative dominance were recorded and are based on basal area; see Table 1. The collective basal area across tenure was 15.5 m². In PPFs, the total basal area was 8.7 m²; with the largest basal area measurements from *P. rigida* (0.43), *Nectandra lanceolata* Nees & Mart. (0.38), and *A. fraxinifolium* (0.25). The total basal area in PAFs measured 6.8 m², with largest measurements from *C. americana* (0.35 and 0.29), *A. columbrina* (0.17), and *Cedrela fissilis* Vell. (0.14).

**Table 1.** Top IVI species for PAFs and PPFs studied, based on percentages of Relative Abundance (R.A.), Relative Dominance (R.D.), and Relative Frequency (R.F.) in six forest types of 0.01 ha under different Land Tenure Regimes (LTR).

| Species | R.A. (%) | Species | R.D. (%) | Species | R.F. (%) | Species | IVI (%) | LTR |
|---|---|---|---|---|---|---|---|---|
| *A. aculeata* | 12.3 | *A. aculeata* | 0.07 | *A. aculeata* | 0.50 | *A. aculeata* | 62 | |
| *A. fraxinifolium* | 7.3 | *C. americana* | 0.04 | *P. rigida* | 0.42 | *P. rigida* | 48 | |
| *C. langsdorfii* | 6.8 | *P. rigida* | 0.02 | *A. fraxinifolium* | 0.38 | *A. fraxinifolium* | 44 | |
| *P. rivularis* | 6.6 | *C. langsdorfii* | 0.02 | *C. americana* | 0.33 | *C. americana* | 37.0 | |
| *P. rigida* | 6.4 | *P. dubium* | 0.02 | *P. dubium* | 0.33 | *P. dubium* | 36 | PAF |
| *C. americana* | 3.6 | *A. columbrina* | 0.01 | *P. rivularis* | 0.29 | *P. rivularis* | 35 | |
| *T. pallida* | 3.6 | *P. rivularis* | 0.01 | *T. pallida* | 0.29 | *T. pallida* | 32 | |
| *R. lorentziana* | 3.0 | *C. fissilis* | 0.01 | *G. ulmifolia* | 0.29 | *G. ulmifolia* | 31 | |
| *A. concolor* | 3.0 | *A. fraxinifolium* | 0.01 | *R. lorentziana* | 0.25 | *R. lorentziana* | 28 | |
| *P. dubium* | 2.7 | *A. urundeuva* | 0.01 | *T. catigua* | 0.25 | *C. langsdorfii* | 27 | |
| *C. retusa* var. *latifolia* | 12.9 | *P. rigida* | 0.04 | *P. rigida* | 0.42 | *O. suaveolens* | 46 | |
| *P. pennatifolius* | 12.7 | *C. retusa var. latifolia* | 0.03 | *O. suaveolens* | 0.42 | *P. rigida* | 46 | |
| *P. dubium* | 5.2 | *C. gonocarpum* | 0.03 | *L. leucanthus* | 0.42 | *L. leucanthus* | 45 | |
| *C. vernalis* | 5.2 | *N. lanceolata* | 0.03 | *C. gonocarpum* | 0.37 | *P. dubium* | 42 | |
| *O. suaveolens* | 4.3 | *O. suaveolens* | 0.03 | *P. dubium* | 0.37 | *C. retusa var. latifolia* | 39 | PPF |
| *Z. petiolare* | 4.3 | *P. dubium* | 0.03 | *Z. petiolare* | 0.32 | *C. gonocarpum* | 39 | |
| *P. rivularis* | 4.1 | *L. leucanthus* | 0.02 | *C. retusa* var. *latifolia* | 0.26 | *Z. petiolare* | 36 | |
| *P. rigida* | 3.9 | *A. fraxinifolium* | 0.02 | *A. fraxinifolium* | 0.26 | *P. pennatifolius* | 34 | |
| *L. leucanthus* | 2.6 | *A. niopoides* | 0.01 | *A. edulis* | 0.26 | *A. fraxinifolium* | 28 | |
| *T. catigua* | 2.6 | *Z. petiolare* | 0.01 | *C. americana* | 0.21 | *A. edulis* | 28 | |

The most abundant species in both land tenure regimes were three understory trees, *P. pennatifolius*, *C. retusa* var. *latifolia*, and *A. aculeata*. In MNCC and MNCK, the savannah-palm forests of *A. aculeata* were dominant at lower elevations, while the *Guazuma ulmifolia* Lam. (Malvaceae), *Trichilia catigua* A. Juss. (Meliaceae), *Allophylus edulis* L. (Sapindaceae), and *Reichenbachia paraguayensis* (D. Parodi) Dugan & Daniel (Nyctaginaceae) were abundant at higher elevations. Within PNY, *C. langsdorfii* was the most abundant. In contrast, the most abundant species in PPFs were: G1, *Cupania vernalis* Cambess. (Sapindaceae) and *P. pennatifolius*; in G2, *C. retusa* var. *latifolia* and *Genipa americana* L. (Rubiaceae); and in JP, *P. rivularis*, *P. rigida*, and *O. suaveolens*. The highest-ranking IVI species of each criterion are presented in Table 1.

Here, SWDI values were between 0.0 and 2.81, presented in Figure 4. The most diverse plots were: G2 plot 10 (2.81), G2 plot 12 (2.43), JP plot 11 (2.41), and MNCC plot 30 (2.24). A score of 0.0 was recorded in MNCC plots 20 and 27 where *A. aculeata* was the only tree species present, as also documented by Steinbrenner, constituting palm-savannah forest types [99]. SWDI mean values between PPFs and PAFs were statistically different based on *t*-test results.

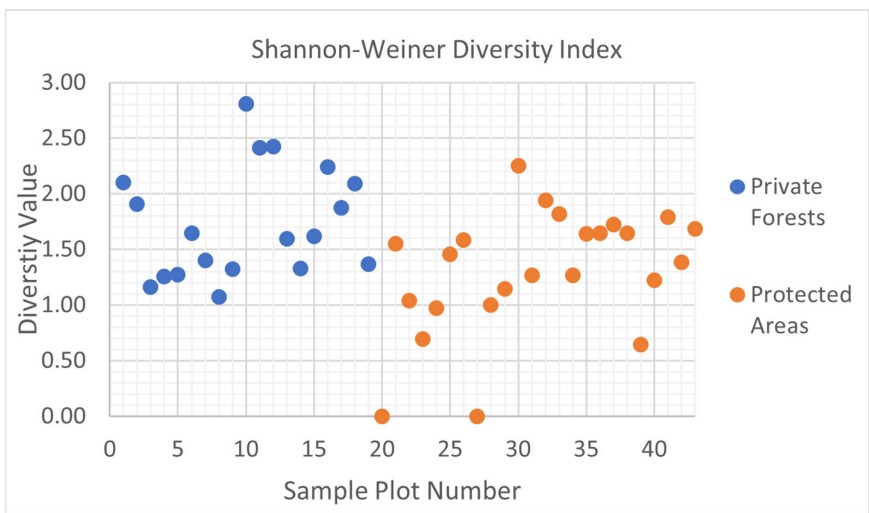

**Figure 4.** Shannon–Weiner Diversity Index values for each plot in study. Private forests ranged from 1.0 to 2.9. Protected areas ranged from 0.0 to 2.3.

## 4. Discussion

As mentioned previously, Paraguay contains the convergence of various ecoregions, thus, species and generalizations across landscapes are difficult to interpret [21,100]. Some authors have characterized the region as neotropical seasonally dry forests [93,101,102], while Spichiger et al., (1995) correlated ecoregion transitions with different species present in both zones and the adaptability of certain species to exist across zones [41]. Here, some site differences were observed at the ecoregion level and not necessarily based on tenure. Most notably, PNY, G1, and G2 are found within the BAAPA ecoregion [5,23,70], where average annual rainfall, topographic undulations, and the presence of iron-rich clay soils increases in an easterly direction as noted by [103,104]; whereas, formations of MNCK, MNCC, and JP were characterized by shallow sandy soils, savannah grassland pockets, rocky hills, and secondary forests of the Humid Chaco ecoregion [5,23,70]. Here, and in the literature, ecosystem characteristics were proximity to water, elevation, slope, soil structure, anthropogenic intervention, influence species distribution, richness, and size [15,30,105–111]. Significant anthropogenic disturbances occurred for decades in MNCC and MNCK through exploitative mining for cobble stone roads (in 1993, both were declared protected areas), compounded with the high proximity to the AMA and large buffer zone populations with cultural identities of forest extraction for use [69]. Hence, both are classified as degraded forests, savannah—palm forests, and vegetation on rocky slopes. Likewise, the waterways and land of PNY were utilized to support Paraguay's first iron foundry. In 1973, this westernmost portion of Atlantic Forest landscape was declared a national park to be conserved. Considering environmental threats observed across land tenure regimes, Guairá PPF sites are not tourist destinations [16], but rather are used as sources of work and income, as forage sites, or for natural resources, yet still subject to potential watershed contamination from the local–international mining industry. The same can be said for JP; however, the surrounding population has left noticeable impacts on forest structure (illegal harvests, forest extractions, and pollution). G1 and G2 also lie within the Atlantic Forest and were in greater proximity to riparian areas—G1 demonstrated gallery forest composition, and G2 sites contained transitions between gallery—riparian forest formations with water-tolerant species, as echoed by authors, Vogt & Mereles, (2005) and others [112–116]. Canopy trees in PPFs were primarily shade tolerant species adapted to deep soiled environments, *L. leucanthus*, *O. suaveolens*, and fruit bearing *C. gonocarpum* [11,13,15,30]. Furthermore, *O. suaveolens* reflect the importance of laurel species in the region, as five genera and 20 species of Lauraceae trees are registered throughout the country [117].

In this study, there was less species richness in PAFs than in PPFs. Differences could be related to species turnover along gradients of humidity, soils, elevation, and disturbance as

documented by [105,106]. Furthermore, natural disturbances like fire undoubtedly affected the distribution, representation, and mensuration of samples. In October 2020, wildland fires heavily burned MNCK and JP. While the JP inventory was completed beforehand, the MNCK inventory data contained fire tolerant species (like *G. ulmifolia* and *A. aculeata*) and exotic genera (*Hovenia, Citrus, Eriobotrya*). Fire, heat, and drought are ecological filters that can quickly change the plant structure communities, as seen also in the Pantanal, the Amazon, and Chaco [118–120]. To which, we cannot examine effects of climate cycles by sampling in a single year since the dynamic under consideration lasts several decades (or longer), and species richness might have rebounded in fire sites given an extended study. Flora, fauna, and ecosystems can adapt and coevolve with human disturbance, selection throughout time, but with accelerated land conversion, niche, threatened, or data deficient species may be lost in silence [121].

Considering *dbh* values between land tenure regimes, *dbh* was not significantly different, and could be due to the rate of succession across all forests. Albeit secondary forests with relatively small mean *dbh*s can still contain different species compositions depending upon regional, genetic, and climatic conditions. The diameter classes in both PPFs and PAFs with the greatest number of individuals was in the 0.01–0.09 m range. Signifying that in both land tenure regimes most trees belonged to the understory or were part of secondary growth forests. Echoed again in the IVI relative dominance results where collective basal area peaked in intermediate *dbh* classes. Except for *P. rigida*, all large diameter samples are late succession species that require certain forest features like canopy shade and soil structure to be established. In sites with large basal area samples, there is increased crown competition favoring shade loving understory species. As noted with highly abundant species in both land tenure regimes, *P. pennatifolius, C. retusa* var. *latifolia*, and *P. rivularis*. While Fabaceae trees *L. leucanthus, P. rigida*, and *P. dubium*, are indicative of secondary forests, and their high frequency in both land tenure regimes agrees with data by [11,13,15,30]. The highest IVI across PAFs was for *A. aculeata* (62.3%), thriving in palm-savannah formations in the Humid Chaco, yet absent entirely from all Atlantic Forest PNY transects, likely to elevation, sunlight, and soil composition differences.

The Shannon–Wiener diversity index measured transect biodiversity by examining abundances of multiple species in each transect. For reference, SWDI values for tropical forests near the equator in Brazil oscillate between 3.83 and 5.85, whereas monoculture plantations have a value of 0.0 [111]. Paraguayan subtropical forests differ from tropical forests due to a greater shift in seasonality and less rainfall throughout the year. Here, SWDI average PPF diversity values were statistically greater than average PAF diversity values based on *t*-test results. Considering this, we must question the efficacy of management styles and conservation efforts, regional climatic conditions, and regional threats to ecosystems. As mentioned above, the G1 and G2 PPF rural sites were less altered by anthropogenic intervention and relatively well conserved, whereas the natural resources of JP, MNCK, MNCC, and PNY have all been culled multiple times. Nonetheless, due to a limited number of sample plots, population sizes, and plot dimensions across different geopolitical departments and ecoregions, no large-scale inferences about forest dynamics or composition between PPFs and PAFs could be made based on this study alone.

This study was conducted as a Peace Corps Masters International environmental conservation volunteer, living and serving in a rural community of a foreign country, Paraguay, where the goal was to learn, share, and support local community. Thus, multiple limiting factors manifested throughout, such as time constraints, transportation, funding, and access to higher levels of academia. To effectively monitor the state of forests across tenure, further studies with additional measurements with stricter spatial and temporal dimensions are required to observe fluctuations in values with natural, anthropogenic disturbances over time. Furthermore, future studies incorporating information of the economic value of forests, restorative forestry, and cross-sectoral collaborations can result in more holistic support for ecosystems and global communities [122].

## 5. Conclusions

This preliminary study compared floristic composition between protected areas and private property forests in eastern Paraguay. The families most abundant were Fabaceae, Rutaceae, Myrtaceae, and Rhamnaceae. The most important species as determined by IVI in PPFs were *O. suaveolens, L. leucanthus,* and *C. gonocarpum*, in PAFs were *A. aculeata, A. fraxinifolium*, and *C. americana*, and in both land tenure regimes, *P. rigida*, and *P. dubium.* There was greater species richness in PPFs than in PAFs. Results of the SWDI demonstrated floristic diversity was statistically higher in PPFs than in PAFs; however, the average *dbh* between both populations was not statistically different. Differences are related to higher diversity primary forest plots located in rural Atlantic Forest and the substantial anthropogenic histories of PAFs sampled, resulting in secondary forests. Results can serve local landowners and rural communities by helping better understand what tree species are present, which can lead to more informed forest management strategies. On a regional level, this study supports the scientific community and public and private sectors with registries of tree populations, plots, and structural, taxonomic, and diversity data of the BAAPA—Humid Chaco gradient in Paraguay. This study also offers regional data from Guairá never studied before and within protected areas: PNY, MNCC, and MNCK, which can serve as reference material for future studies and encourage further cross-sectoral collaborative investigation and management of rural and protected area sites. Globally, this study adds to the interconnected understanding of forests and biodiversity in Paraguay, South America. Future studies must continue to be conducted in Paraguay to support forest restoration, natural and cultural resource stability for biodiversity, and millions of forest-dependent people.

**Author Contributions:** Conceptualization, A.G.C. and S.C.R.; methodology, A.G.C., P.T.V.G. and S.C.R.; investigation, A.G.C.; writing—original draft preparation, A.G.C. writing—review and editing, A.G.C. and S.C.R.; supervision, S.C.R.; project administration, S.C.R.; funding acquisition, S.C.R. All authors have read and agreed to the published version of the manuscript.

**Funding:** This research received no external funding.

**Institutional Review Board Statement:** Not applicable.

**Informed Consent Statement:** Not applicable.

**Data Availability Statement:** The data presented in this study are available on request from the corresponding author.

**Acknowledgments:** The author acknowledges the ecology, flora, and wonderful people living in Paraguay. Gratitude is expressed to the families of San Jose, Oñondivepa, Ñu Vera, San Agustin, and Mangrullo that allowed investigation to occur in their forests. Thanks is offered to MADES, Park Guards, and staff that granted permission to study in the protected areas, and those who supported field work. Much gratitude is expressed to the Oregon State University College of Forestry, and Seri C. Robinson, the Peace Corps Masters International Program, Peace Corps Paraguay Environmental Conservation Sector, and friends and family who supported this study throughout.

**Conflicts of Interest:** The authors declare no conflict of interest.

## Appendix A

**Table A1.** Plot characteristics of all sites. Abbreviations: Guairá Region 1 (G1), Guairá Region 2 (G2), Jacare Piru (JP), Natural Monument Cerro Koi (MNCK), Natural Monument Cerro Chorori (MNCC), National Park Ybycuí (PNY); Ecoregions: Atlantic Forest (AF), Humid Chaco (CH).

| Site | Plot No. | Mgmt. | Eco Region | Coordinates | Ele. (m) | Gnd. Cvr. (%) | Tot. Area (m$^2$) | Scale Factor |
|---|---|---|---|---|---|---|---|---|
| G1-SJ1 | 1 | PF | AF | 25°41.426′ S 55°59.140′ W | 264.0 | 75 | 100 | 0.0059 |
| G1-SJ2 | 2 | PF | AF | 25°41.419′ S 55°59.135′ W | 262.1 | 85 | 100 | |

**Table A1.** *Cont.*

| Site | Plot No. | Mgmt. | Eco Region | Coordinates | Ele. (m) | Gnd. Cvr. (%) | Tot. Area (m²) | Scale Factor |
|------|----------|-------|------------|-------------|----------|----------------|-----------------|--------------|
| G1-SJ3 | 3 | PF | AF | 25°41.435′ S 55°59.129′ W | 259.4 | 95 | 100 | |
| G1-Oño1 | 4 | PF | AF | 25°41.962′ S 56° 20.145′ W | 283.4 | 80 | 100 | |
| G1-Oño2 | 5 | PF | AF | 25°41.273′ S 56° 13.460′ W | 232.6 | 65 | 100 | |
| G1-Oño3 | 6 | PF | AF | 25°41.362′ S 56° 24.440′ W | 235.3 | 75 | 100 | |
| G2-SA1 | 7 | PF | AF | 25°50.334′ S 55°52.597′ W | 242.9 | 90 | 100 | 0.0248 |
| G2-SA2 | 8 | PF | AF | 25°50.862′ S 55°52.967′ W | 243.2 | 70 | 100 | |
| G2-SA3 | 9 | PF | AF | 25°50.809′ S 55°52.988′ W | 242.3 | 60 | 100 | |
| G2-NV1 | 10 | PF | AF | 25°50.486′ S 56° 02.316′ W | 210.9 | 45 | 100 | |
| G2-NV2 | 11 | PF | AF | 25°50.027′ S 56°02.061′ W | 196.9 | 50 | 100 | |
| G2-NV3 | 12 | PF | AF | 25°50.058′ S 56°02.019′ W | 198.4 | 70 | 100 | |
| JP 1 | 13 | PF | CH | 25°19.632′ S 57°22.809′ W | 110.3 | 65 | 100 | 0.0011 |
| JP 2 | 14 | PF | CH | 25°19.582′ S 57°22.623′ W | 130.5 | 70 | 100 | |
| JP 3 | 15 | PF | CH | 25°19.839′ S 57°22.884′ W | 144.8 | 50 | 100 | |
| JP 4 | 16 | PF | CH | 25°19.672′ S 57°22.795′ W | 99.7 | 83 | 100 | |
| JP 5 | 17 | PF | CH | 25°19.556′ S 57°22.585′ W | 109.7 | 55 | 100 | |
| JP 6 | 18 | PF | CH | 25°19.532′ S 57°22.587′ W | 127.7 | 80 | 100 | |
| JP 7 | 19 | PF | CH | 25°19.826′ S 57°22.968′ W | 134.1 | 85 | 100 | |
| MNCK 1 | 20 | PA | CH | 25°19.474′ S 57°23.762′ W | 176.8 | 40 | 100 | 0.045 |
| MNCK 2 | 21 | PA | CH | 25°19.543′ S 57°23.807′ W | 167.3 | 65 | 100 | |
| MNCK 3 | 22 | PA | CH | 25°19.485′ S 57°23.839′ W | 209.1 | 70 | 100 | |
| MNCK 4 | 23 | PA | CH | 25°19.437′ S 57°23.916′ W | 141.1 | 90 | 100 | |
| MNCK 5 | 24 | PA | CH | 25°19.341′ S 57°24.020′ W | 192.9 | 83 | 100 | |
| MNCK 6 | 25 | PA | CH | 25°19.332′ S 57°24.049′ W | 192.6 | 90 | 100 | |
| MNCK 7 | 26 | PA | CH | 25°19.311′ S 57°24.079′ W | 198.7 | 97 | 100 | |
| MNCK 8 | 27 | PA | CH | 25°18.806′ S 57°24.212′ W | 192.9 | 100 | 100 | |
| MNCK 9 | 28 | PA | CH | 25°19.439′ S 57°23.903′ W | 633 ft | 70 | 100 | |
| MNCC 1 | 29 | PA | CH | 25°18.837′ S 57°24.105′ W | 168.3 | 55 | 100 | 0.018 |
| MNCC 2 | 30 | PA | CH | 25°18.880′ S 57°24.153′ W | 167.3 | 40 | 100 | |
| MNCC 3 | 31 | PA | CH | 25°18.873′ S 57°24.134′ W | 159.1 | 60 | 100 | |
| MNCC 4 | 32 | PA | CH | 25°18.811′ S 57°24.212′ W | 199.6 | 80 | 100 | |
| MNCC 5 | 33 | PA | CH | 25°18.838′ S 57°24.103′ W | 160.0 | 50 | 100 | |
| MNCC 6 | 34 | PA | CH | 25°18.832′ S 57°24.257′ W | 167.3 | 60 | 100 | |
| MNCC 7 | 35 | PA | CH | 25°18.785′ S 57°24.129′ W | 168.9 | 33 | 100 | |

**Table A1.** *Cont.*

| Site | Plot No. | Mgmt. | Eco Region | Coordinates | Ele. (m) | Gnd. Cvr. (%) | Tot. Area (m²) | Scale Factor |
|------|----------|-------|------------|-------------|----------|---------------|----------------|--------------|
| MNCC 8 | 36 | PA | CH | 25°18.786′ S 57°24.211′ W | 188.1 | 75 | 100 | |
| MNCC 9 | 37 | PA | CH | 25°18.787′ S 57°24.210′ W | 182.9 | 63 | 100 | |
| PNY 1 | 38 | PA | AF | 26°05.574′ S 56°50.771′ W | 198.7 | 73 | 100 | 0.000012 |
| PNY 2 | 39 | PA | AF | 26°03.121′ S 56°52.181′ W | 275.5 | 40 | 100 | |
| PNY 3 | 40 | PA | AF | 26°02.252′ S 56°51.687′ W | 363.6 | 92 | 100 | |
| PNY 4 | 41 | PA | AF | 26°05.473′ S 56°50.282′ W | 119.8 | 80 | 100 | |
| PNY 5 | 42 | PA | AF | 26°04.077′ S 56°50.938′ W | 195.4 | 75 | 100 | |
| PNY 6 | 43 | PA | AF | 26°04.423′ S 56°50.615′ W | 222.5 | 50 | 100 | |

**Table A2.** Quantity of samples registered in all sites.

| Private Property Forests | | | |
|---|---|---|---|
| **Family** | **Genera** | **Species** | **Quantity** |
| Rhamnaceae | *Columbrina* | *Columbrina retusa* var. *latifolia* | 60 |
| Rutaceae | *Pilocarpus* | *Pilocarpus pennatifolius* | 59 |
| Fabaceae | *Peltophorum* | *Peltophorum dubium* | 24 |
| Sapindaceae | *Cupania* | *Cupania vernalis* | 24 |
| Lauraceae | *Ocotea* | *Ocotea suaveolens* | 20 |
| Rutaceae | *Zanthoxylum* | *Zanthoxylum petiolare* | 20 |
| Myrtaceae | *Plinia* | *Plinia rivularis* | 19 |
| Fabaceae | *Parapiptadenia* | *Parapiptadenia rigida* | 18 |
| Fabaceae | *Lonchocarpus* | *Lonchocarpus leucanthus* | 12 |
| Meliaceae | *Trichilia* | *Trichilia catigua* | 12 |
| Salicaceae | *Casearia* | *Casearia sylvestris* | 12 |
| Sapotaceae | *Chrysophyllum* | *Chrysophyllum gonocarpum* | 11 |
| Tiliaceae | *Luehea* | *Luehea divaricata* | 9 |
| Anacardiaceae | *Astronium* | *Astronium fraxinifolium* | 7 |
| Moraceae | *Ficus* | *Ficus* sp. | 7 |
| Rubiaceae | *Genipa* | *Genipa americana* | 7 |
| Sapindaceae | *Allophylus* | *Allophylus edulis* | 7 |
| Arecaceae | *Syagrus* | *Syagrus romanzoffiana* | 6 |
| Clusiaceae | *Rheedia* | *Rheedia brasiliensis* | 6 |
| Fabaceae | *Myrocarpus* | *Myrocarpus frondosus* | 6 |
| Myrtaceae | *Plinia* | *Plinia peruviana* | 6 |
| Sapindaceae | *Diplokeleba* | *Diplokeleba floribunda* | 6 |
| Boraginaceae | *Cordia* | *Cordia americana* | 5 |
| Meliaceae | *Trichilia* | *Trichilia pallida* | 5 |
| Myrsinaceae | *Rapanea* | *Rapanea lorentziana* | 5 |
| Sapotaceae | *Chrysophyllum* | *Chrysophyllum marginatum* | 5 |
| Fabaceae | *Anadenanthera* | *Anadenanthera columbrina* | 4 |
| Myrtaceae | *Hexachlyamys* | *Hexachlyamys edulis* | 4 |
| Apocynaceae | *Tabernaemontana* | *Tabernaemontana australis* | 3 |
| Boraginaceae | *Cordia* | *Cordia ecalyculata* | 3 |
| Fabaceae | *Albizia* | *Albizia niopoides* | 3 |
| Lauraceae | *Ocotea* | *Ocotea disopyrifolia* | 3 |
| Lauraceae | *Nectrandra* | *Nectandralanceolata* | 3 |
| Loganiaceae | *Strychnos* | *Strychnos brasiliensis* | 3 |
| Myrtaceae | *Campomanesia* | *Campomanesia xanthocarpa* | 3 |

**Table A2.** *Cont.*

| Private Property Forests | | | |
|---|---|---|---|
| **Family** | **Genera** | **Species** | **Quantity** |
| Polygonaceae | *Ruprechita* | *Ruprechita laxiflora* | 3 |
| Rhamnaceae | *Hovenia* | *Hovenia dulcis* | 3 |
| Rutaceae | *Citrus* | *Citrus aurantium* | 3 |
| Sapindaceae | *Diatenopteryx* | *Diatenopteryx sorbifolia* | 3 |
| Verbenaceae | *Vitex* | *Vitex megapotamica* | 3 |
| Aquifoliaceae | *Ilex* | *Ilex paraguariensis* | 2 |
| Euphorbiaceae | *Actinostemon* | *Actinostemon concolor* | 2 |
| Fabaceae | *Machaerium* | *Machaerium acuifolium* | 2 |
| Fabaceae | *Gleditsia* | *Gleditsia amorphoides* | 2 |
| Fabaceae | *Copaifera* | *Copaifera langsdorfii* | 2 |
| Moraceae | *Cecropia* | *Cecropia pachystachya* | 2 |
| Myrtaceae | *Eugenia* | *Eugenia uniflora* | 2 |
| Rutaceae | *Balfourrodendron* | *Balfourrodendron riedelianum* | 2 |
| Symplocaceae | *Symplocos* | *Symplocos* sp. | 2 |
| Ulmaceae | *Celtis* | *Celtis pubescens* | 2 |
| Arecaceae | *Acrocomia* | *Acrocomia aculeata* | 1 |
| Bignoniaceae | *Handroanthus* | *Handroanthus heptaphylla* | 1 |
| Boraginaceae | *Cordia* | *Cordia trichotma* | 1 |
| Cactaceae | *Cereus* | *Cereus stenogonus* | 1 |
| Cactaceae | *Praecereus* | *Praecereus euchlorus* | 1 |
| Caricaceae | *Jacaratia* | *Jacaratia spinosa* | 1 |
| Euphorbiaceae | *Manihot* | *Manihot grahamii* | 1 |
| Fabaceae | *Enterlobium* | *Enterlobium contrortosiliquum* | 1 |
| Fabaceae | *Pterogyne* | *Pterogyne nitens* | 1 |
| Fabaceae | *Calliandra* | *Calliandra tweediei* | 1 |
| Fabaceae | *Inga* | *Inga uruguensis* | 1 |
| Lauraceae | *Ocotea* | *Ocotea puberula* | 1 |
| Meliaceae | *Cedrela* | *Cedrela fissilis* | 1 |
| Meliaceae | *Cabralea* | *Cabralea canjerana* | 1 |
| Moraceae | *Chlorophora* | *Chlorophora tinctoria* | 1 |
| Myrtaceae | *Capparicordis* | *Capparicordis tweediana* | 1 |
| Rhamnaceae | *Rhamnidium* | *Rhamnidium elaeocarpum* | 1 |
| Rutaceae | *Helietta* | *Helietta apiculata* | 1 |
| Rutaceae | *Fagara* | *Fagara naranjillo* | 1 |
| Sapindaceae | *Melicoccus* | *Melicoccus lepidopetalus* | 1 |
| Sapindaceae | *Diatenopteryx* | *Diatenopteryx sorbifolia* | 1 |
| Urticaceae | *Urera* | *Urera baccifera* | 1 |
| Protected Area Forests | | | |
| Family | Genera | Species | Quantity |
| Arecaceae | *Acrocomia* | *Acrocomia aculeata* | 54 |
| Anacardiaceae | *Astronium* | *Astronium fraxinifolium* | 32 |
| Fabaceae | *Copaifera* | *Copaifera langsdorfii* | 30 |
| Myrtaceae | *Plinia* | *Plinia rivularis* | 29 |
| Fabaceae | *Parapiptadenia* | *Parapiptadenia rigida* | 28 |
| Meliaceae | *Trichilia* | *Trichilia pallida* | 16 |
| Boraginaceae | *Patagonula* | *Patagonula americana* | 15 |
| Euphorbiaceae | *Actinostemon* | *Actinostemon concolor* | 13 |
| Myrsinaceae | *Rapanea* | *Rapanea lorentziana* | 13 |
| Rutaceae | *Helietta* | *Helietta apiculata* | 12 |
| Fabaceae | *Peltophorum* | *Peltophorum dubium* | 12 |
| Clusiaceae | *Rheedia* | *Rheedia brasiliensis* | 11 |
| Sapotaceae | *Chrysophyllum* | *Chrysophyllum gonocarpum* | 10 |
| Malvaceae | *Guazuma* | *Guazuma ulmifolia* | 10 |
| Fabaceae | *Anadenanthera* | *Anadenanthera columbrina* | 9 |
| Meliaceae | *Trichilia* | *Trichilia catigua* | 9 |
| Sapindaceae | *Diatenopteryx* | *Diatenopteryx sobifolia* | 8 |

**Table A2.** *Cont.*

| Family | Genera | Species | Quantity |
|---|---|---|---|
| **Private Property Forests** | | | |
| Lauraceae | *Ocotea* | *Ocotea suaveolens* | 8 |
| Nyctaginaceae | *Rychenbachia* | *Rychenbachia paraguayensis* | 8 |
| Anacardiaceae | *Astronium* | *Astronium urundeuva* | 7 |
| Rubiaceae | *Genipa* | *Genipa americana* | 6 |
| Fabaceae | *Albizia* | *Albizia hassleri* | 5 |
| Moraceae | *Ficus* | *Ficus enormis* | 5 |
| Fabaceae | *Holocalyx* | *Holocalyx balansae* | 5 |
| Tiliaceae | *Luehea* | *Luehea divaricata* | 5 |
| Myrtaceae | *Myrciaria* | *Myrciaria cuspidata* | 5 |
| Meliaceae | *Cedrela* | *Cedrela fissilis* | 4 |
| Ulmaceae | *Celtis* | *Celtis pubescens* | 4 |
| Rutaceae | *Citrus* | *Citrus aurantium* | 4 |
| Sapindaceae | *Cupania* | *Cupania vernalis* | 4 |
| Fabaceae | *Machaerium* | *Machaerium paraguariense* | 4 |
| Rutaceae | *Pilocarpus* | *Pilocarpus pennatifolius* | 4 |
| Nyctaginaceae | *Pisonia* | *Pisonia aculeata* | 4 |
| Salicaceae | *Casearia* | *Casearia gossypiosperma* | 3 |
| Moraceae | *Cecropia* | *Cecropia pachystachya* | 3 |
| Myrtaceae | *Hexachlamys* | *Hexachlamys edulis* | 3 |
| Myrtaceae | *Psidium* | *Psidium guayaba* | 3 |
| Rhamnaceae | *Rhamnidium* | *Rhamnidium elaeocarpum* | 3 |
| Annonaceae | *Rollinia* | *Rollinia emarginata* | 3 |
| Cannabaceae | *Trema* | *Trema micrantha* | 3 |
| Meliaceae | *Cabralea* | *Cabralea canjerana* | 2 |
| Rubiaceae | *Coutarea* | *Coutarea hexandra* | 2 |
| Lauraceae | *Nectandra* | *Nectandra lanceolata* | 2 |
| Piperaceae | *Piper* | *Piper amalago* | 2 |
| Sapotaceae | *Sideroxylon* | *Sideroxylon obtusifolium* | 2 |
| Sapindaceae | *Allophylus* | *Allophylus edulis* | 1 |
| Myrtaceae | *Campomanesia* | *Campomanesia xanthocarpa* | 1 |
| Cactaceae | *Cereus* | *Cereus stenogonus* | 1 |
| Boraginaceae | *Cordia* | *Cordia americana* | 1 |
| Boraginaceae | *Cordia* | *Cordia ecalyculata* | 1 |
| Boraginaceae | *Cordia* | *Cordia trichotma* | 1 |
| Rutaceae | *Fagara* | *Fagara rhoifolia* | 1 |
| Binoniaceae | *Handroanthus* | *Handroanthus impetiginosus* | 1 |
| Malvaceae | *Heliocarpus* | *Heliocarpus americanus* subsp. *Popayanensis* | 1 |
| Bignoniaceae | *Hexandra* | *Hexandra heptaphylla* | 1 |
| Fabaceae | *Inga* | *Inga uruguensis* | 1 |
| Anacardiaceae | *Lithraea* | *Lithraea molleoides* | 1 |
| Myrtaceae | *Plinia* | *Plinia peruviana* | 1 |
| Lauraceae | *Ocotea* | *Ocotea diospyrifolia* | 1 |
| Euphorbiaceae | *Sapium* | *Sapium hematospermum* | 1 |
| Moraceae | *Sorocea* | *Sorocea bonplandii* | 1 |
| Arecaceae | *Syagrus* | *Syagrus romanzoffiana* | 1 |
| Annonaceae | *Xylopia* | *Xylopia brasiliensis* | 1 |

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
