# Peer review of "Forest Inventories in Private and Protected Areas of Paraguay"

_challenges, doi:10.3390/challe14020023_

Round 1

Reviewer 1 Report

Dear authors,

I am glad to read and to be a part as a review of such great work. Here a some of my observations and suggestions for your paper:

-Introduction and Background are too extensive. Please try to shorten and leave out the unnecessary.

-Line 103: Plese find error in the brackets.

-Paraguay contains the convergence of various ecoregions and generalizations across its landscapes are difficult to interpret. So, you have to summarize the most important results and measurements. Please, in the conclusion, add the most important results in terms of the presence of families and species and measured parameters (dbh) in private  forests and protected areas.

Author Response

Thank you very much for reviewing our manuscript. Please see attachment below. 

Reviewer 2 Report

This study is focus on a relevant topic within biodiversity and conservation knowledge areas. The findings of this study are highly important for defining of Subtropical forest inventories as well as delineating proper forest management for each studied site. Undoubtedly, this study showed vast fieldwork and gave a clear example of a collaborative and symbiotic work. Some major changes must be done in the manuscript. In general, the authors should be more precise in the information given, trying to select the information related to the objectives and the methodology used to the fulfillment of them.

Overall observation:

The term “tenure” seems to be not specific for the related knowledge areas. I suggest using “land tenure regimes” all over the manuscript.

Abstract in English:

-line 14:

Change “This preliminary study investigated comparisons and differences between..” by “This preliminary study determined the differences and similarities between..”

-line 20: avoid using abbreviation in the abstract (see “dbh”)

The information given in the last sentences is not related to the findings of this study; please try to be more precise.

Abstract in English:

The coherence and grammatical structure of this section must be corrected by a Spanish speaking person. Many mistakes have been detected. Please, avoid using literal translation.

Introduction:

This section is too long (almost 5 pages). Despite all the information given is highly relevant, not all of this information is related to the studied issues. I suggest grouping the information into two sub-sections: 1. Biological, climatic and landscape characterization of the studied area. 2. Status of forest management practices in both land tenure regimes (highlighting those relevant for the assessed parameters). After these, introduce the objectives of the study.

The quality of Figures 1, 2 and 3 are very poor. The information written in these figures is not legible. I suggest combining all these figures in one including the political borders of Paraguay, precipitation distribution and location of the studied sites.

The last paragraph of this section should be included in the Discussion (maybe it needs to be re-organized within that section).

Materials and Methods:

-line 241:

After “by FAO” add “with some modifications.”

-line 247:

Please, indicate which “ecological indicator” was used. Please clarify where all this information is summarized (lines 246-248).

-line 253:

Please, indicate in the Reference section the whole URL of this website.

-line 264:

Please, note that the first time that “dbh” appears should be fully mentioned and then always use the abbreviation.

-lines 283-284 and lines 316-318: should be incorporated in a subsection “Data Analysis”, where the statistical software used also must be mentioned.

-Equation in line 280, change “b” by “dbh”

-Lines 295-297. Please, indicate the measurement units for all the parameters

-Line 304. Please, indicate the names of the indexes at the beginning.

-The subsection “future studies” should be moved to the Discussion section.

Results and Discussion:

The information given in this section should be reorganized. 

-All the information related to transect establishment in each studied site (in both land tenure regimes) must be moved to M&M.

-In the first sub-section (3.1), only information related to the characterization of each studied site should be given.

-Combine Figure 1 and 2. Left (a) to PPF and Right (b) to PAF. Use the same colors for the legend and put only one legend. Please, enlarge the pie charts. Eliminate the title, this information must be given in the caption figure in the bottom.

-After line 417 should be discussed the possible causes of the obtained results. 

-The sentences in lines 440-444 should be moved to line 432.

-In the third subsection: Be careful, some sentences are repetitive and related to the M&M section. For example, lines: 455-458, 459, 474, etc. 

-Line 470: what is the meaning of “across management”, is it referred to one of one of the tenure regimes? It could be confusing

-Figures 6, 7, 8 and 9 could be combined in one figure: left to PFP (a) and right to PAF (b). The titles and the numbers of individuals characterized must be mentioned in the caption figure. Do not mention total basal area in the caption figure, these results could be mentioned in the text directly. 

-Combine Table 1 and 2 in only one Table. Add a new column at the left where information related to land tenure regimes must be mentioned. 

-Line 514-515: change “was not a surprise” by “it is in agreement with the data reported by…”.

-Line 532-533: Please, indicate explicitly which kind of tenure was superior and discuss possible causes. 

Conclusions:

Most of the information given in this section is not direct of the findings of this study.  Authors use a narrative of the issue too general. The conclusion section must be restricted to the relevance of the information generated through this study, to whom and why this information is useful and it is possible to add some suggestions for future studies. 

Acknowledgements:

It would be interesting, if it were possible, that some of the name of Paraguayans association, institutions or groups that collaborated with the authors to fulfill the objectives of this work would be mentioned here.

Author Response

Thank you very much for reviewing and recommending numerous ways in which we can improve our manuscript. Please see the attachment below. 

Thank you,

Round 2

Reviewer 1 Report

Dear authors,

Thank you for accepted suggestions for improving your work.

Author Response

Thank you for reviewing and editing our manuscript. Please see the attachment. 

Reviewer 2 Report

I really appreciate the changes done by the authors. Now, the manuscript is well organized and more fluent when reading it. Moreover, the information selected for the Introduction section is more adequate for the studied topics and to understand the objectives, results and discussion of this study. 

Nevertheless, some minor changes should be taken into account before the acceptance of the final manuscript version.

Abstract in English:

line 8: change “relate” by “relates”

line 20: change “a statistically significant” by “significantly”

line 22: change “variations” by “detected differences” 

Abstract in Spanish:

Although a significant translation improvement was done, some grammatical problems remain. 

line 27: change “existe a través de” by “está relacionado con”

line 28: after “y” add “se”

line 28: change “con los incendios y sequía” by “con el aumento de la frecuencia de incendios y sequía”

Change the sentence: “El gobierno ha tomado pasos para mitigar la deforestación, como leyes, parcelación de tierras privadas, y establecimiento de áreas protegidas federales” by “ El gobierno local ha decidido actuar para mitigar la deforestación a través de la promulgación de leyes, parcelación de tierras privadas, y establecimiento de áreas protegidas federales”

line 29: change “sigue como pregunta” by “siguen siendo desconocidas”

line 29: change “símiles” by “similitudes”

line 33: change  “tres bosques familiares privadas” by “tres bosques de gestión privada a cargo de familias”

line 34: reorder the sentence as follow: “Se establecieron 43 transectas siguiendo los protocolos…” 

line 36: reorder the sentence as follow “Se registraron 903 muestras, las cuales pertenecieron a 35 familias, …”

line 40: change “cuentan” by “muestran”

line 40: change “y estadísticamente significativa mayor” by “significativamente mayor”

line 41: change “estadísticamente significativa mayor” by “significativamente mayor”

line 41: change “DAP” by “diameter” 

line 41: change “en tierras privadas que en áreas” by “en las tierras privadas respecto a las áreas”

line 42: change” las variaciones eran” by “las diferencias encontradas fueron”

Introduction:

line 59: add “some” before “regions”

line 61: eliminate “Integral to planetary health,”

Change the sentence (line 65-67) by “The country is officially bilingual, including Guaraní and Spanish languages, and contains 19 indigenous populations from…” 

Figure 1: the map on the upper right corner is not legible, I suggest eliminating it. Please, indicate which sites correspond to PPF and which to PAF, for example clarifying the used color for the sites´names.

line 164: eliminate the sentences: “This is a challenge”

lines 180-187: the information given in these sentences are mentioned in M&M. I suggest eliminating them. 

M&M

line 191: be careful, 14 or 17 geopolitical departments? Inconsistency between the information given in the Introduction and in M$M sections. 

line 192: change “are found” by “are located”

line 202: eliminate “PPF”

line 203: add “located at” before “32 km”

line 247: eliminate “Only”

Results:

Figure 2: the names in x axes are not legible

Add “Plant” before “Families” in the caption figure

Figure 3: the names in x axes are not legible

eliminate “Representation of”

line 344: change “High frequency samples in PPFs follow:” by “High frequency species in PPFs were:”

line 353-354: where is recorded the information of absolute and relative dominance?

Discussion:

line 383: add “thus,” before “species”

line 396: add “in” before “MNCC”

line 434: add “with” before “the greatest”

line 443: eliminate the repeated “in”

line 455: eliminate “well”

line 460: change”are made based on” by “could be made based on”

Author Response

(The authors gave the same response as above.)
